# Transforming carbon dioxide into a methanol surrogate using modular transition metal-free Zintl ions

Bono van IJzendoorn [1], Saad F. Albawardi [1], William D. Jobbins[2], George F. S. Whitehead [2], John E. McGrady [1]✉ & Meera Mehta [1]✉

Although not the only greenhouse gas, $CO_2$ is the poster child. Unsurprisingly, therefore, there is global interest across industrial and academic research in its removal and subsequent valorisation, including to methanol and its surrogates. Although difficult to study, the heterogenous pnictogens represent one important category of catalytic materials for these conversions; their high crustal abundance and low cost offers advantages in terms of sustainability. Here, Zintl clusters based on these elements are studied as homogenous atom-precise models in $CO_2$ reduction. A family of group 13 functionalized pnictogen clusters with the general formula $[(R_2E)Pn_7]^{2-}$ (E = B, Al, In; Pn = P, As) is synthesized and their catalytic competency in the reduction of $CO_2$ probed. Trends in both turnover numbers and frequencies are compared across this series, and $[(iBu_2Al)P_7]^{2-}$ found to be very high-performing and recyclable. Electronic structures across the series are compared using density functional theory to provide mechanistic insights.

Carbon dioxide ($CO_2$) is perhaps the most notorious and well-recognized greenhouse gas. Its reputation is well earned, as there has been almost a 50% increase in atmospheric $CO_2$ since the industrial revolution, which is directly linked to global warming, associated natural disasters, rising sea levels and ice sheet degradation[1–3]. This increase in atmospheric carbon is a direct result of global energy dependence on fossil fuel combustion. Methanol ($CH_3OH$) is one alternative clean fuel source, as well as a basic C1 building block for the molecules that underpin everyday life, from pharmaceuticals and agrochemicals to adhesives, paints and coatings[4,5]. Globally, 140 million metric tonnes of $CH_3OH$ were consumed in 2018, and this demand is expected to double by 2030[6]. Strategies that recycle $CO_2$ directly into methanol (or its surrogates), which can then be burned for energy or used as industry feedstocks, offer the possibility of closing the carbon cycle[3,7–10], and it is therefore no surprise that facilitating this transformation is of great global interest. However, the high thermo-dynamic stability of $CO_2$ means that technologies that mediate its transformation often rely on catalysts featuring transition metals, including those that fall in the Platinum Group[11–16]. The high cost of these metals is an obvious impediment to their widespread application, and inexpensive and sustainable alternatives are an important target[7].

One possible approach to minimize costs is to replace the expensive transition metal catalysts with inexpensive ones based on earth-abundant elements[17–20], where systems based on heterogenous phosphorus are of increasing interest[21–24]. Although their structures are not yet fully understood, red and violet phosphorus contain clusters of phosphorus between four and nine units that are polymerized and crosslinked together[25–27]. Heterogenous catalysts are often more robust and easier to separate compared to homogenous systems, minimizing purification costs and enabling catalyst recycling[28]. In situ mechanistic insights will be essential to release the full catalytic potential of materials based on heterogenous phosphorus, but these studies are extremely difficult to perform due to their extreme insolubility. In this context, Zintl clusters based on the $[P_7]$ architecture offer an alternative source of mechanistic insight. Clusters of this type can be considered as intermediates between mono-nuclear phosphorus species and bulk solids, and they have the advantage of being

[1]Department of Chemistry, University of Oxford, 12 Mansfield Road, Oxford OX1 3QR, UK. [2]Department of Chemistry, University of Manchester, Oxford Road, Manchester M13 9PL, UK. ✉e-mail: John.mcgrady@chem.ox.ac.uk; Meera.mehta@chem.ox.ac.uk

easily synthetically accessible, soluble in common solvents, and they provide convenient spectroscopic handles to probe reactivity. A detailed understanding of catalytic activity of [P₇] clusters may, therefore, offer a window into the analogous chemistry with heterogenous phosphorus.

The application of Zintl clusters as components in catalyst design is a new field[29], and in most of the previous reports, the clusters act as innocent ligands that support catalytically active transition metals (Fig. 1). For example, Goicoechea and Weller have coordinated [Ge₉] to Rh metal and mediated the hydrogenation of olefins, and then later used the same system in H/D exchange reactions (Fig. 1)[30,31]. Fässler and co-workers have also employed a [Ge₉] cluster, in this case with Ni coordinated to mediate alkene isomerization[32]. Although not derived from a Zintl species, Scheschkewitz and co-workers also facilitated alkene isomerization with a silicon-iridium cluster system[33]. Sun and co-workers used Ru encapsulated by a Sn cluster dispersed on a CeO₂ surface to affect the reverse water-gas shift reaction[34], although the extent to which the cluster remains intact after dispersion is still unclear.

In 2022, we reported that the transition metal-free boron-functionalized [P₇] cluster [κ²-(BBN)P₇]²⁻ ([1]²⁻; BBN = 9-borabicyclo[3.3.1] nonane) was catalytically competent for the hydroborative reduction of C=O bonds in carbonyls and CO₂, as well as the C=N bonds in isocyanates, carbodiimides, imines, pyridines and nitriles[35,36]. In this paper, we extend that work by preparing a wider family of group 13 functionalized heptapnictogen clusters with the general formula [κ²-(R₂E)Pn₇]²⁻ (E = B, Al, In, Pn = P, As). Whilst the [κ²-(Ph₂In)P₇]²⁻ cluster has previously been prepared by the Goicoechea group[37], the synthesis of clusters [κ²-(iBu₂Al)P₇]²⁻ and [κ²-(iBu₂Al)As₇]²⁻ are reported here. We compare the catalytic performance of this series in the hydroborative reduction of $CO_2$ gas with respect to selectivity, turnover number (TON), and turnover frequency (TOF), and show that this family of catalysts are highly competent with the aluminium species being the most active. By systematically modifying catalyst design, we are able to better understand their reactivity landscape, and in this work, the identity of the group 13 moiety is found to have a major impact on reactivity. Our mechanistic studies further allowed us to uncover a catalytic off-cycle intermediate, and consider an alternative mechanism for this transformation. Additionally high catalyst recyclability is proven, an added advantage of employing systems based on functionalized [Pn₇] clusters.

## Results and discussion

### Synthesis and properties of [κ²-(R₂E)Pn₇]²⁻ clusters

The [κ²-(iBu₂Al)Pn]²⁻ (Pn = P ([2]²⁻), As ([3]²⁻)) clusters were synthesized using protocols analogous to [κ²-(BBN)P₇]²⁻ ([1]²⁻)[35]; specifically diisobutylaluminium hydride was dehydrocoupled with the protonated clusters [HPn₇]²⁻ (Pn = P, As)[38,39], leading to H₂ gas evolution (Fig. 2a). In line with the nuclear magnetic resonance (NMR) spectroscopic features reported for [κ²-(R₂E)P₇]²⁻ (R₂E = BBN, Ph₂In)[35,37], the [κ²-(iBu₂Al)P₇]²⁻ cluster reveals five resonances in the ³¹P NMR spectrum consistent with κ²-coordination to the [P₇] core. Multiple efforts were made to acquire a ²⁷Al NMR spectrum to establish the coordination mode of Al to the phosphorus cluster, but with no success. The synthesis of the related [κ²-((Me₂N)₂Ga)P₇]²⁻ was also investigated by reacting (Me₂N)₃Ga dimer with [HP₇]²⁻ and resulted in ³¹P NMR spectra consistent with the formation of the expected product. However, despite multiple efforts isolation of analytically pure material was not achieved and precluded further catalytic investigations (see Supplementary Information section 2.3).

Single crystals were acquired for both the [Na(18c6)]₂[2] and [K(18c6)]₂[3] (18c6 = 18-crown-6) salts and X-ray diffraction (XRD) studies confirmed the κ²-coordination mode of the Al to the [Pn₇] core, see Fig. 2b, c. The bond metric data obtained by the XRD analysis allows for direct comparison between the [κ²-(iBu₂Al)Pn₇]²⁻ clusters, and hence, the influence of the [Pn₇] core on the substitution. Firstly, as expected, the average Al−P bonds in [2]²⁻ (2.4429(14) Å) are shorter compared to the average Al−As bonds in [3]²⁻ (2.528(19) Å) due to the larger atomic radius of As compared to P[40]. Also consistent with the atomic size difference between phosphorus and arsenic, the P−Al−P bond angle in [2]²⁻ is more acute (84.46(3)°) when compared to the As−Al−As bond angle in [3]²⁻ (89.2(4)°). The bond distances within the [Pn₇] cores are similar to those in the unfunctionalized [Pn₇]³⁻ clusters, and differences are again consistent with the differences in atomic radius of As and P[41]. Meaningful comparisons can also be made between the family of group 13 functionalized phosphorus clusters [κ²-(R₂E)P₇]²⁻ (E = B, Al, In) clusters[35,37], where the average E−P bond lengths increase down the group (B−P: 2.067(11) Å in [1]²⁻; Al−P: 2.4429(14) Å in [2]²⁻; In−P: 2.578(2) Å in [4]²⁻). Further, the P−E−P bond angles become more acute for the heavier group 13 elements (P−B−P: 93.3(3)° in [1]²⁻, P−Al−P: 84.46(3)° in [2]²⁻, P−In−P: 80.33(4)° in [4]²⁻): presumably this changing bond angle reflects the approximately constant bite angle of the P₇ unit leading to more acute angles as the bonds get longer.

### Catalytic reduction of carbon dioxide

We have previously reported a detailed study of the catalytic reduction of carbonyl functional groups using [1]²⁻[35], and clusters [2]²⁻ to [4]²⁻ all showed similar reactivity towards acetophenone and benzaldehyde that is consistent with activation of the unsaturated C=O bond (see Supplementary Information section 4.1). Motivated by this observation, the catalytic competence of the family of clusters [κ²-(R₂E)Pn₇]²⁻ (E = B, Al, In, Pn = P, As) ([1]²⁻ - [4]²⁻) towards the hydroborative reduction of CO₂ was probed using the previously reported optimised conditions: 1 atm of CO₂, 0.33 % catalyst loading; room temperature

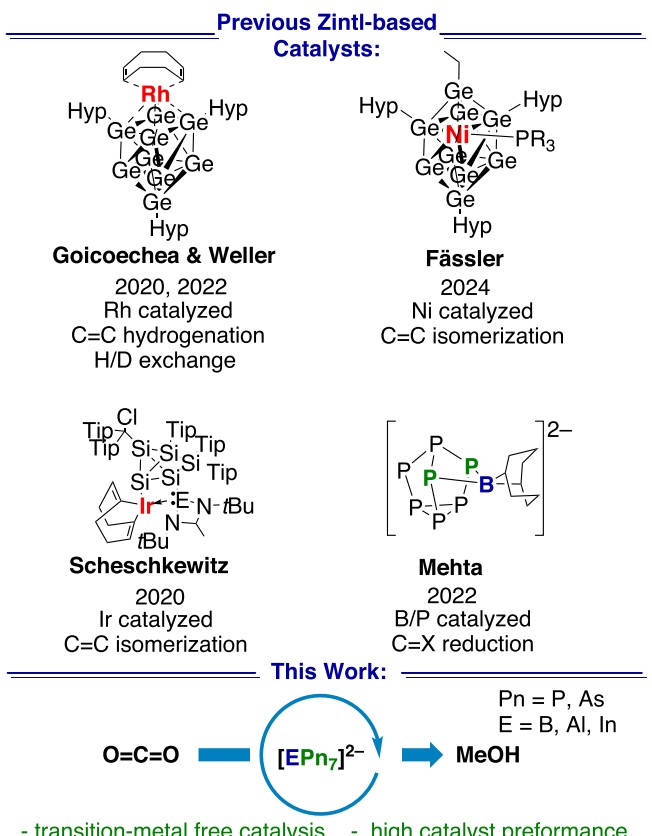

**Previous Zintl-based Catalysts:**

**Goicoechea & Weller**
2020, 2022
Rh catalyzed
C=C hydrogenation
H/D exchange

**Fässler**
2024
Ni catalyzed
C=C isomerization

**Scheschkewitz**
2020
Ir catalyzed
C=C isomerization

**Mehta**
2022
B/P catalyzed
C=X reduction

**This Work:**

O=C=O → [EPn₇]²⁻ → MeOH

Pn = P, As
E = B, Al, In

- transition-metal free catalysis
- new family of catalysts
- high catalyst preformance
- mechanistic understanding

**Fig. 1 | Select examples of Zintl clusters that have been applied in homogenous catalysis, and this work.** Hyp = Si(SiMe₃)₃, Tip = 2,4,6-triisopropylphenyl, E = Si, Ge, or Sn.

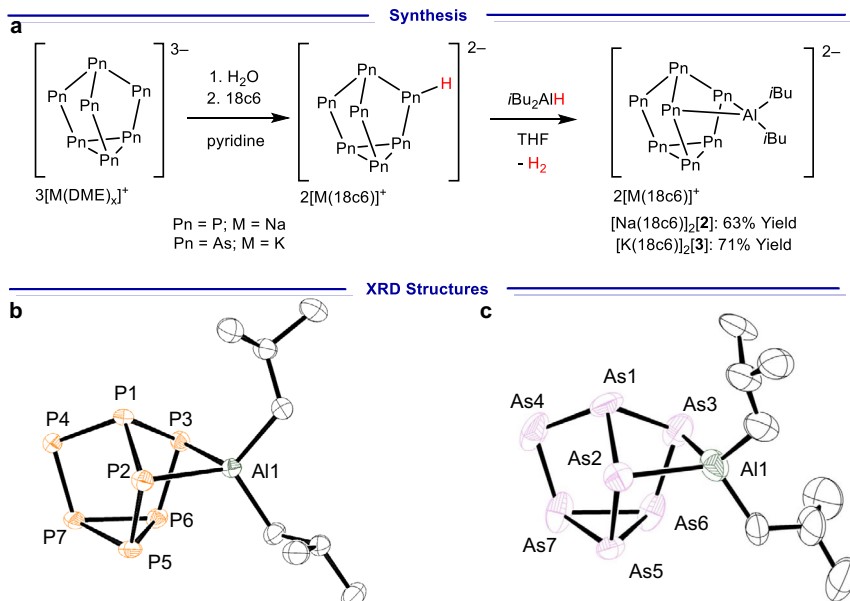

**Fig. 2 | Synthesis and X-ray diffraction structures of [Na(18c6)]₂[2] and [K(18c6)]₂[3]. a** Synthesis of [Na(18c6)]₂[2] and [K(18c6)]₂[3]. 18c6 = 18-crown-6. **b** Molecular structure of [κ²-(iBu₂Al)P₇]²⁻ ([2]²⁻) in the [Na(18c6)]₂[(iBu₂Al)P₇] salt. Anisotropic displacement ellipsoids pictured at 50% probability. Hydrogen atoms, and [Na(18c6)]⁺ counter cations omitted for clarity. Phosphorus: orange; aluminium: green; carbon: white. Selected bond lengths [Å]: Al1–P2 2.4385(10), Al1–P3 2.4472(10), P1–P2 2.127(10), P1–P3 2.1877(10), P1–P4 2.1592(10), P2–P5 2.1809(10), P3–P6 2.1807(10), P4–P7 2.1433(10), P5–P6 2.2589(11), P5–P7 2.2346(10), P6–P7

2.2338(10); selected bond angle [°]: P2–Al1–P3 84.46(3). **c** Molecular structure of [κ²-(iBu₂Al)As₇]²⁻ ([3]²⁻) in the [K(18c6)]₂[(iBu₂Al)As₇] salt, two. Anisotropic displacement ellipsoids pictured at 50% probability. Hydrogen atoms, and [K(18c6)]⁺ counter cations omitted for clarity. Arsenic: plum; aluminium: green; carbon: white. Selected bond lengths [Å]: Al1–As2 2.513(9), Al1–As3 2.529(9), As1–As2 2.460(5), As1–As3 2.423(4), As1–As4 2.352(5), As2–As5 2.383(4), As3–As6 2.380(4), As4–As7 2.340(6), As5–As6 2.469(4), As5–As7 2.467(6), As6–As7 2.464(4); selected bond angle [°]: As2–Al1–As3 89.7(3).

(RT); 18 h; o-difluorobenzene (oDFB):toluene (Tol) solvent system; and HBBN dimer as the reductant. The findings are summarized below in Table 1. We note that we were only able to prepare the Na⁺ salt of [2]²⁻ and the K⁺ salt of [3]²⁻. However, for [4]²⁻, where both the Na⁺ and K⁺ salts could be prepared, and there is no discernible difference in TOF_RT. This observation is fully consistent with our previous findings for [κ²-(BBN)P₇]²⁻ ([1]²⁻) where both the Na and K salts were also tested[35], and suggests that the identity of the s-block cation does not influence catalytic performance to any great extent. Isotopic labelled studies using ¹³C-labelled CO₂ confirmed the origin of the carbon products (see Supplementary Information section 3.2.).

Taking the catalytic performance of [Na(18c6)]₂[1] as a baseline (TOF_RT = 30 h⁻¹, TOF_50 °C = 300 h⁻¹, 99% selectivity for MeOBBN [a methanol surrogate which can be quantitatively and easily hydrolyzed to methanol[35]]), the aluminium-functionalized cluster [2]²⁻ displayed greater reactivity at 0.33 mol% loading both at RT (TOF_RT = 38 h⁻¹) and at 50 °C (TOF_50 °C = 577 h⁻¹). The TOF_50 °C for the Al cluster [2]²⁻ is almost twice that of the boron cluster [1]²⁻, but the selectivity decreases to 65% MeOBBN, with the remaining 34% of product being the diol surrogate CH₂(OBBN)₂. Selectivity towards MeOBBN can be recovered by reducing the catalyst loading from 0.33 mol% to 0.10 mol% at RT without any great impact on TOF_RT (42 h⁻¹). For the As analogue, [κ²-(iBu₂Al)As₇]²⁻ ([3]²⁻), slightly higher TOFs are observed at 0.33 mol% loading when compared to the [κ²-(BBN)P₇]²⁻ ([1]²⁻) (TOF_RT = 33 h⁻¹ and TOF_50 °C = 361 h⁻¹). The selectivity towards MeOBBN is again lower than for [κ²-(BBN)P₇]²⁻ ([1]²⁻), and again it can be increased by reducing the catalyst loading to 0.10 mol% with no significant impact on TOF_RT (34 h⁻¹). The indium-functionalized cluster [κ²-(Ph₂In)P₇]²⁻ ([4]²⁻) was found to have poorer catalytic performance when compared to the other [κ²-(R₂E)Pn₇]²⁻ systems investigated here, with maximum TOF_RT of 13 h⁻¹ and TOF_50 °C of 156 h⁻¹. As measured by TOF, the catalytic performance across the entire series decreases in the order: [κ²-(iBu₂Al)P₇]²⁻ > [κ²-(iBu₂Al)As₇]²⁻ > [κ²-(BBN)P₇]²⁻ > [κ²-(Ph₂In)P₇]²⁻.

In order to probe the origins of the lower selectivity of [2]²⁻ and [3]²⁻ for MeOBBN compared to [1]²⁻, the reaction mixtures were monitored by ¹H NMR spectroscopy. The reaction profiles for the hydroboration of CO₂ using 0.33 mol% [κ²-(iBu₂Al)P₇]²⁻ or [κ²-(iBu₂Al) As₇]²⁻ as catalyst at RT are given in Supplementary Information Figs. S14 and S15, and both show initial rapid formation of CH₂(OBBN)₂ which then decays over time to form MeOBBN. Unlike the corresponding reaction profiles, we previously reported for catalyst [1]²⁻[35], not all the CH₂(OBBN)₂ is fully consumed before the remaining HBBN dimer has all reacted (observed by ¹H NMR spectroscopy). Neither the carboxyl-BBN (HC(=O)OBBN) nor formaldehyde is observed under these conditions, in line with literature reports where it is believed that the susceptibility towards hydroboration follows the order HC(=O) OBBN > CO₂ > CH₂(OBBN)₂ > MeOBBN, with HC(=O)OBBN being the most reactive[42]. We propose that the more polarized Al–Pn (Pn = P, As) bonds (when compared to the B–P bonds), in combination with the relatively high oxophilicity and electropositivity of Al compared to other group 13 elements[43], makes catalysts [2]²⁻ and [3]²⁻ more prone to CO₂ capture. As CO₂ hydroboration to MeOBBN requires three hydroboration steps, activation of a new molecule of CO₂ competes with activation of a previously hydroborated CH₂(OBBN)₂ molecule for re-entry into the catalytic cycle. We also noticed that selectivity for MeOBBN increases at lower catalyst loadings; we believe this is related to the amount of catalytic material at the solution-gas interface, where the concentration of CO₂ is highest. To test this concept, a dilution experiment was conducted where catalyst loading relative to borane was held constant, but an overall increase in solvent would also result in less catalytic material at the solvent-atmosphere interface (see Supplementary Information section 3.3.2). In Table 1 the 0.33 mol% catalyst loading [κ²-(iBu₂Al)P₇]²⁻ reaction at RT was performed at 1.94 mM catalyst concentration and resulted in a 35:65 product distribution of CH₂(OBBN)₂:MeOBBN, but dilution to 0.97 mM catalyst concentration now resulted in a 25:75 product distribution. Another approach to change the amount of catalytic material at the solution-

**Table 1 | Hydroboration of $CO_2$ by [$\kappa^2$-(R$_2$E)Pn$_7$]$^{2-}$ clusters**

$$CO_2 + 1.5\ (HBBN)_2 \xrightarrow[RT]{Cat,\ oDFB/Tol} HC(=O)OBBN + CH_2(OBBN)_2$$
$$\mathbf{a} \qquad\qquad \mathbf{b} \qquad\qquad \mathbf{c}$$
$$+ H_3COBBN + CH_4 + (BBN)_2O$$
$$\mathbf{d} \qquad \mathbf{e}$$

| Cat | Loading (%)[a] | Time (h) | Temp. (°C) | b Conv. (%)[b] | c Conv. (%)[b] | d Conv. (%)[b] | e Conv. (%)[b] | TOF (h⁻¹)[b] |
|---|---|---|---|---|---|---|---|---|
| [K(18c6)]₂ **[1]**[35] | 0.33 | 10 | RT | 0 | 0 | >99 | 0 | 30 |
| [Na(18c6)]₂ **[1]**[35] | 0.33 | 10 | RT | 0 | 0 | >99 | 0 | 30 |
| | 0.33 | 1 | 50 | 0 | 0 | >99 | 0 | 300 |
| | 0.10 | 32 | RT | 0 | 0 | >99 | 0 | 31 |
| [Na(18c6)]₂ **[2]** | 0.33 | 8 | RT | 0 | 34 | 65 | 0 | 38 |
| | 0.33 | 0.52 | 50 | 0 | 34 | 65 | 0 | 577 |
| | 0.10 | 24 | RT | 0 | 0 | >99 | 0 | 42 |
| [K(18c6)]₂ **[3]** | 0.33 | 9.1 | RT | 0 | 30 | 69 | 0 | 33 |
| | 0.33 | 0.83 | 50 | 0 | 29 | 70 | 0 | 361 |
| | 0.10 | 30 | RT | 0 | 0 | >99 | 0 | 34 |
| [K(18c6)]₂ **[4]** | 0.33 | 24 | RT | 0 | 0 | >99 | 0 | 13 |
| [Na(18c6)]₂ **[4]** | 0.33 | 24 | RT | 0 | 0 | >99 | 0 | 13 |
| | 0.33 | 1.92 | 50 | 0 | 10 | 89 | 0 | 156 |
| | 0.10 | 120 | RT | 0 | 0 | 98 | 0 | 8 |

[a]Relative to B–H bonds.
[b]Determined by ¹H NMR spectroscopy, based on C–H bond formation using hexamethylbenzene as an internal standard.

gas interface, would be to change the diameter ($\varnothing$) of the reaction vessel (see Supplementary Information section 3.3.2). To this end, when the hydroboration of $CO_2$ (0.33 mol% loading [$\kappa^2$-($iBu_2Al$)$P_7$]$^{2-}$, RT) was repeated again at 1.94 mM concentration but performed in a J Young ampoule ($\varnothing = 30$ mm) instead of in a J Young NMR tube ($\varnothing = 5$ mm) the ratio of products was found to be 63:37 $CH_2(OBBN)_2$:MeOBBN product distribution. Both of these experiments are consistent with the local concentration of catalyst at the solution-gas interface having an impact on product selectivity, where less catalytic material correlates to greater MeOBBN selectivity.

Since catalyst [Na(18c6)]$_2$[$\kappa^2$-($iBu_2Al$)$P_7$] ([Na(18c6)]$_2$[**2**]) was found to be the most active in terms of TOF, efforts were made to optimize the reaction conditions to select for the formation of MeOBBN (Table 2). High catalyst loadings of 10% and 1% (entries 1 and 2) showed low selectivity for MeOBBN, with only approximately 50% conversion. Further, at 10 mol% catalyst loading crystals of [Na(18c6)] [(HCO$_2$)$_2$BBN] (compound **28b'** in ref. 35) could be collected[35]. In the previous discussion of the data in Table 1 we demonstrated that reducing the catalyst loading to 0.1 mol% at RT increases selectivity, with MeOBBN then being observed as the sole product. In an effort to increase TOF, 0.1 mol% catalyst loading of [**2**]$^{2-}$ was investigated at 50, 60 and 70 °C (entries 3–5) but, in these cases, selectivity towards MeOBBN decreased and at 70 °C overall conversion also decreased. Heating of [Na(18c6)]$_2$[$\kappa^2$-($iBu_2Al$)$P_7$] revealed rapid catalyst decomposition at 70 °C, slower decomposition at 60 °C, and no decomposition at 50 °C, as measured by $^{31}P$ NMR spectroscopy (see Supplementary Information section 3.3.3). Lowering the catalyst loading further to 0.05 mol% at RT (entry 6) revealed a decrease in TOF$_{RT}$ while retaining high selectivity and a high TON of 2000. Increasing the temperature with 0.05 mol% loading (entries 7–9), again decreased selectivity. Lowering the loading even further to 0.01 mol%, (entries 10 and 11) gave the highest TONs of 9400 and 9600 at RT and 50 °C, respectively. For [Na(18c6)]$_2$[$\kappa^2$-($iBu_2Al$)$P_7$] ([Na(18c6)]$_2$[**2**]) a maximum TOF of 741 h$^{-1}$ considering all products and 600 h$^{-1}$ considering only MeOBBN was obtained. Analysis of these TON and TOFs reveal that [Na(18c6)]$_2$[$\kappa^2$-($iBu_2Al$)$P_7$] ([Na(18c6)]$_2$[**2**]) outperforms our previously reported boron-functionalized analogue [**1**]$^{2-}$ in its ability to mediate this transformation, and in fact, it is highly competitive with other main group catalysts and even many homogenous transition metal catalysts (see Supplementary Information sections 3.6. and 3.7.).

## Recycling

Previously, we reported that [Na(18c6)]$_2$[$\kappa^2$-(BBN)$P_7$] ([Na(18c6)]$_2$[**1**]) could be recycled 7 times in the catalytic hydroboration of $CO_2$ with no loss in performance[35]. To investigate the recyclability of the [Na(18c6)]$_2$[$\kappa^2$-($iBu_2Al$)$P_7$] ([Na(18c6)]$_2$[**2**]) catalyst, the hydroboration of $CO_2$ was performed using 0.1 mol% loading at RT and after complete conversion investigation by $^{31}P$ NMR spectroscopy revealed that the catalyst was still present (see Supplementary Information Fig. S29). Furthermore, minimal changes in catalytic competency were observed after reloading the reaction mixture with HBBN dimer and $CO_2$ 9 times, indicating living catalysis (see Fig. 3). After these cycles, the catalyst was recovered from the reaction mixture and re-used in an independent reaction batch (cycle 11 Fig. 3), and again no decrease in catalytic performance was observed. Overall these experiments show that the [Na(18c6)]$_2$[$\kappa^2$-($iBu_2Al$)$P_7$] ([Na(18c6)]$_2$[**2**]) catalyst is very robust with a total of 3.18 mmol MeOBBN being produced from 0.3 µmol [**2**]$^{2-}$, a 10778-fold excess. This type of high recyclability amongst homogenous catalysts is uncommon, and highlights one of the advantages of building systems based on the [Pn$_7$] framework.

## Mechanistic investigations

Benzaldehyde was used as a model substrate to investigate the kinetics of hydroboration as it can be easily weighed into the reaction mixture and, unlike gaseous $CO_2$, primarily resides in the solution phase with the other reagents. Mimicking conditions used in the hydroboration of $CO_2$ (0.1 mol% loading, RT, 1:1 $o$DFB:Tol), variable time normalization analysis was applied to probe the order of the reagents (data regarding VTNA analysis available in Supplementary Data 2). Both the [Na(18c6)]$_2$[**1**] and [Na(18c6)]$_2$[**2**] catalysts were investigated following the VTNA method described by the Burés group (see Supplementary Information section 5), where the reaction order can be obtained via a graphical representation[44,45]. For both [Na(18c6)]$_2$[**1**] and [Na(18c6)]$_2$[**2**] catalysts using HBBN dimer as reductant, the analysis supports a fitting of a zero-order in the concentration of aldehyde and catalyst and a half-order in the concentration of HBBN dimer. This suggests that the rate limiting step is a reaction involving HBBN dimer but not the catalysts, therefore we propose the breaking of the HBBN dimer is rate limiting. When changing the reductant from HBBN dimer to the monomeric HBpin, the analysis supports a fitting of a zero-order in the concentration of aldehyde and the first order in the concentration of borane and in the concentration of catalysts. These findings suggest a fast activation of the C=O bond and slow activation of the B–H bond, consistent with the observed stoichiometric reactivity discussed below. To further investigate the potential rate-determining step, kinetic isotope effects were probed by performing the hydroboration of benzaldehyde with DBpin (0.1 mol% loading, RT). The initial rate from the DBpin reaction was determined and compared to the initial rate of the analogous reaction with HBpin. A kinetic isotope effect ($K_H/K_D$) was found at 2.16, a value that is in good agreement with other hydroboration catalysts where B–H activation or hydride transfer is presumed to be one of the slow steps[46–48].

In our previous report on the catalytic hydroboration of $CO_2$ by [Na(18c6)]$_2$[$\kappa^2$-(BBN)$P_7$] ([Na(18c6)]$_2$[**1**]), we also performed a complementary series of 1:1 stoichiometric reactions with a range of unsaturated molecules including benzaldehyde, acetophenone and phenyl isocyanate, and also with the boranes HBpin and HBBN dimer[35]. In that study, we noted that [Na(18c6)]$_2$[$\kappa^2$-(BBN)$P_7$] reacts rapidly with C=O bonds but rather more slowly with B–H bonds. Here, the corresponding reactivity of the Al analogue [Na(18c6)]$_2$[$\kappa^2$-($iBu_2Al$)$P_7$] ([Na(18c6)]$_2$[**2**]) towards the same set of substrates is probed. A 1:1 reaction of [Na(18c6)]$_2$[**2**] with HBpin at RT revealed no reaction (monitored using $^{11}B$ and $^{31}P$ NMR spectroscopy), precisely as was observed for [Na(18c6)]$_2$[**1**]. However, when the HBBN dimer was used in place of HBpin, a new cluster was identified after 10 hours with $^{11}B$ and $^{31}P$ NMR resonances that are consistent with the insertion of HBBN into one of the Al–P bonds of [Na(18c6)]$_2$[**2**]. In contrast, the stoichiometric reaction of [Na(18c6)]$_2$[**2**] with benzaldehyde at RT revealed almost immediate formation of two other clusters by $^{31}P$ NMR spectroscopy, along with some unreacted [Na(18c6)]$_2$[**2**] (Fig. 4a). Addition of a second equivalent of benzaldehyde drove the reaction further, such that only one of the two products is observed. The $^{31}P$ NMR spectrum of the final product has five peaks in a 2:1:1:1:2 integral ratio, consistent with a symmetric [$P_7$] cluster system with a mirror plane, similar to [Na(18c6)]$_2$[**2**] itself. On that basis, the two new clusters are proposed to be the mono-inserted benzaldehyde ([$\kappa^2$-($iBu_2Al$-O(Ph)(H)C)$P_7$]$^{2-}$, [**5**]$^{2-}$) and bis-inserted benzaldehyde ([$\kappa^2$-($iBu_2Al$-(O(Ph)(H)C)$_2$)$P_7$]$^{2-}$, [**6**]$^{2-}$) products shown in Fig. 4a. $^1H$ NMR, $^{13}C$ NMR, $^{31}P$ COSY NMR, and $^1H^{13}C$ HSQC (Heteronuclear Single Quantum Coherence) NMR spectroscopic studies are fully consistent with this proposal (see Supplementary Information section 4.1.3). When the starting material [**2**]$^{2-}$ was added to a reaction mixture containing only the bis-inserted product [**6**]$^{2-}$, over several hours, the mono-inserted product [**5**]$^{2-}$ was detected by $^{31}P$ NMR spectroscopy, consistent with an equilibrium between [**5**]$^{2-}$ and [**6**]$^{2-}$. If the bulkier acetophenone is used in place of benzaldehyde, only the mono-inserted ([$\kappa^2$-($iBu_2Al$-O(Ph)(Me)C)$P_7$], [**7**]$^{2-}$) is observed: no evidence for the bis-inserted product was detected by $^{31}P$ NMR spectroscopy. The mono-inserted products have a distinctive 7 resonance pattern in the $^{31}P$ NMR spectrum, and the most downfield chemical shift ($^{31}P$ $\delta$: 115 ppm) is similar

**Table 2 | Hydroboration of $CO_2$ with catalyst [2]$^{2-}$**

$$CO_2 + 1.5 \ (HBBN)_2 \xrightarrow[\text{oDFB/Tol, Time, Temp.}]{[Na(18c6)]_2[2] \ (X \ mol\%)} HC(=O)OBBN + CH_2(OBBN)_2$$
$$(1 \ atm) \qquad\qquad + H_3COBBN + (BBN)_2O$$

| Entry | [Na(18c6)]$_2$[2] (mol %)[a] | Time (h) | T (°C) | MeOBBN Conv. (%)[b] | CH$_2$(OBBN)$_2$ Conv. (%)[b] | TON[c] | TOF (h$^{-1}$)[c] |
|---|---|---|---|---|---|---|---|
| 1 | 10 | 1 | RT | 51 | 0 | 5 | 5 |
| 2 | 1.0 | 3 | RT | 58 | 39 | 97(58) | 33(19) |
| 3 | 0.10 | 1.75 | 50 | 80 | 19 | 1000(800) | 571(457) |
| 4 | 0.10 | 1.35 | 60 | 81 | 18 | 1000(810) | 741(600) |
| 5 | 0.10 | 2 | 70 | 50 | 5 | 550(500) | 224(250) |
| 6 | 0.05 | 72 | RT | >99 | 0 | 2000 | 28 |
| 7 | 0.05 | 7.5 | 40 | 50 | 25 | 1500 (1000) | 200(133) |
| 8 | 0.05 | 4.3 | 50 | 80 | 19 | 2000 (1600) | 465(372) |
| 9 | 0.05 | 3 | 60 | 80 | 2 | 1760 (1600) | 586(533) |
| 10 | 0.01 | 400 | RT | 94 | 0 | 9400 | 24 |
| 11 | 0.01 | 66 | 50 | 96 | 0 | 9600 | 145 |

No carboxyl-borane or methane formation was observed by $^1$H NMR spectroscopy.
[a]Relative to B–H bonds.
[b]Determined by $^1$H NMR spectroscopy, based on C–H bond formation using hexamethylbenzene as an internal standard.
[c]TON and TOF considering only MeOBBN formation given in parentheses.

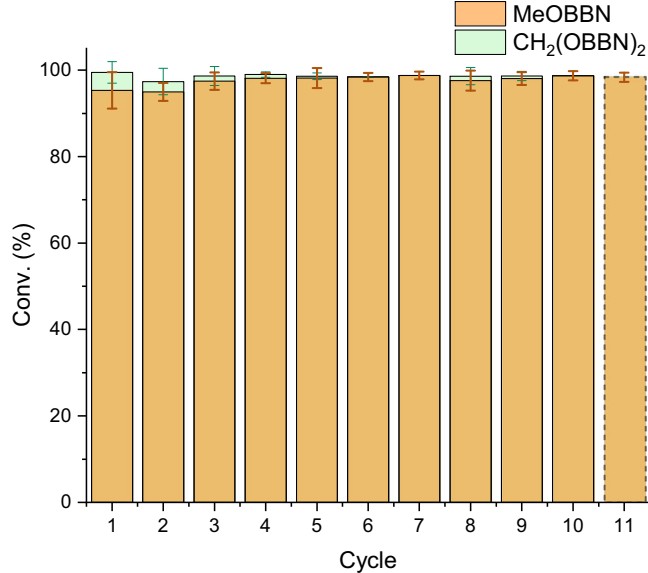

**Fig. 3 | Catalyst recycling in the hydroboration of CO₂ using 0.1 mol% loading [Na(18c6)]₂[2].** Cycle 11 is after the catalyst was recovered from cycle 10 and added to a fresh batch of the reaction. Error bars indicate the 95% confidence interval.

to other P–C carbonyl functionalized clusters[49,50], indicating $[P_7]$–C bond formation. The formation of the bis-inserted product may be precluded with increasing steric bulk at the carbonyl. It is worth noting here that in the case of $CO_2$ activation, steric bulk is minimal, and thus, bis-insertion to give structures analogous to [6]²⁻ should be possible.

Stoichiometric reductions of benzaldehyde were then performed using [Na(18c6)]₂[κ²-($i$Bu₂Al)P₇] ([Na(18c6)]₂[2]) and HBpin as the hydride source (Fig. 4c). Initially, [Na(18c6)]₂[2] was allowed to react with 1 equivalent of benzaldehyde, as described above, and subsequently 1 equivalent of HBpin was added; the Bpin-hydroborated product was obtained in 8% yield, and no transfer of the $i$Bu₂Al-unit to the carbonyl was observed. ³¹P NMR spectroscopy confirmed the presence of [5]²⁻ and [6]²⁻ in the reaction mixture with small residual amounts of [2]²⁻. The order in which the reagents are added has a strong influence on the yield; if the order of addition is reversed so that we add 1 equivalent HBpin first to [Na(18c6)]₂[κ²-($i$Bu₂Al)P₇] followed by addition of 1 equivalent benzaldehyde, the hydroborated product is generated in 58% yield. ³¹P NMR spectra show, in this case, that the dominant species in solution is [2]²⁻ with only minor amounts [5]²⁻ and [6]²⁻. These observations are consistent with the formation of the bis-inserted [6]²⁻ product being an off-cycle thermodynamic sink which inhibits the formation of hydroborated product. However, when the reductant is present, hydroboration of the C=O bond competes with the formation of the bis-inserted product, and thus, the hydroborated product is observed in higher yield. Related reactions were also conducted with [Na(18c6)]₂[κ²-(BBN)P₇] ([Na(18c6)]₂[1]), [K(18c6)]₂[κ²-($i$Bu₂Al)As₇] ([K(18c6)]₂[3]) and [Na(18c6)]₂[κ²-(Ph₂In)P₇] ([Na(18c6)]₂[4]), and again it was found that the order of addition influenced the observed conversion to hydroborated product with the difference being significantly less pronounced for [4]²⁻ (Fig. 4b–e). In our previous work, when [1]²⁻ was studied in a related reaction with acetophenone it was found that there is a 5% exchange between the boron moieties on the cluster and reducing agent[35] and here (Fig. 4b) a similar exchange is observed with benzaldehyde. Whereas with the Al and In systems no such exchange between the group 13 moieties is observed.

In our previous paper, we explored a range of possible intermediates for the reaction of [1]²⁻ with a model carbonyl compound, $H_2C=O$[35]. Here, we extend these studies to include the Al and In

analogues using the more realistic benzaldehyde as a model, with the aim of identifying key features that underpin the differences in reactivity observed in the previous section. Specifically, we focus on the reactions of the clusters with the benzaldehyde, and explore how differences in the nature of the E–Pn and E–O bonds influence the ability of the cluster to bind carbonyl functionalities. In the computational experiments we use Me to model the $i$Bu groups on the Al centres, but the ligands are otherwise as described in the crystallographic data (all calculated coordinates can be found in Supplementary Data 1). Selected bond lengths and Gibbs energies for the addition of one and two equivalents of benzaldehyde to [1]²⁻, [(Me₂Al)P₇]²⁻, [(Me₂Al)As₇]²⁻, [(Me₂Ga)P₇]²⁻ and [4]²⁻ are summarized in Table 3. The values of ΔG shown in the Table correspond to the sequential free energies of benzaldehyde binding: [X]²⁻ + C₆H₅CHO→[X]²⁻•C₆H₅CHO (first row for each cluster) and [X]²⁻•C₆H₅CHO + C₆H₅CHO→[X]²⁻•2C₆H₅CHO (second row). The Pn–E and Pn–O bond lengths are also summarized in Table 3. We note that the approximate $C_S$ symmetry of the double adduct [2]²⁻•2C₆H₅CHO, is fully consistent with the 2:1:1:1:2 ratio noted in the NMR experiments of [6]²⁻. In each case the binding of C₆H₅CHO involves the formation of new Pn–C and E–O bonds with simultaneous cleavage of a Pn–E bond and the π component of a C=O double bond. If we regard the Pn–C and C=O double bonds as approximately constant (at least for the P₇-containing clusters), the trends in free energy should then reflect the difference between the bond energies of the E–P and E–O bonds. Binding of the first molecule of benzaldehyde to the aluminium cluster [(Me₂Al)P₇]²⁻ is clearly thermodynamically favourable (more so than the corresponding reaction with the boron cluster, [1]²⁻), while the free energy for binding the second is close to zero, consistent with [5]²⁻ and [6]²⁻ being in equilibrium. The change of pnictogen, from P to As ([(Me₂Al)P₇]²⁻ to [(Me₂Al)As₇]²⁻), has negligible impact on the equilibria, but the heavier group 13 element In binds benzaldehyde much less strongly with both reactions in Table 3 being endergonic. The relative affinities of the different clusters for benzaldehyde appear, therefore, to correlate with catalytic activity for $CO_2$ reduction shown in Table 1 (Al > B > In). The structural trends in the mono-adducts offer some insight into the origin of these striking differences in energetics. If we compare the Al adduct to its In analogue, the E–P bond length increases from 2.45 Å to 2.60 Å (Δ$r$ = 0.15 Å), while the E–O bond length increases from 1.79 Å to 2.12 Å (Δ$r$ = 0.33 Å). This pattern suggests that the strength of the E–O bond is affected more by the change from E = Al to E = In than is the E–P bond, and this rather low affinity of indium for the hard alkoxide ligand (a well-established trend in p-block chemistry) leads to the weaker binding of benzaldehyde in the case of [4]²⁻. The greater oxophilicity of the Al-functionalized cluster compared to the indium analogue is consistent with the tendency to bind two molecules of benzaldehyde observed in the experiments, and also with the strong dependence of product distribution on the order of addition observed for the Al systems but not for In. Although a Ga-functionalized cluster could not be cleanly synthetically accessed and tested, the computed benzaldehyde binding affinities for [(Me₂Ga)P₇]²⁻ suggest that its catalytic competency should be intermediate between those of the In- and Al-functionalized systems.

The mono-benzaldehyde adduct [2]²⁻•C₆H₅CHO (identified as [5]²⁻ in the NMR experiments) is an aluminium alkoxide species, and there have been numerous computational investigations of the catalytic performance of such species in borohydride reduction of carbonyl groups. Li and co-workers have explored the HBpin hydroboration of $CO_2$ using a range of aluminium hydride species, based on a cycle of C=O insertion into an Al–H bond to form a formyl species followed by trans-metallation to regenerate the hydride along with the product, HCO₂Bpin[51]. By adopting the same fundamental steps, we propose a possible mechanism starting from [(Me₂Al)P₇]²⁻ in Fig. 5, the full free-energy surface for the same pathway is shown in the supplementary information, Fig. S119. At the level of theory used here, the overall free-

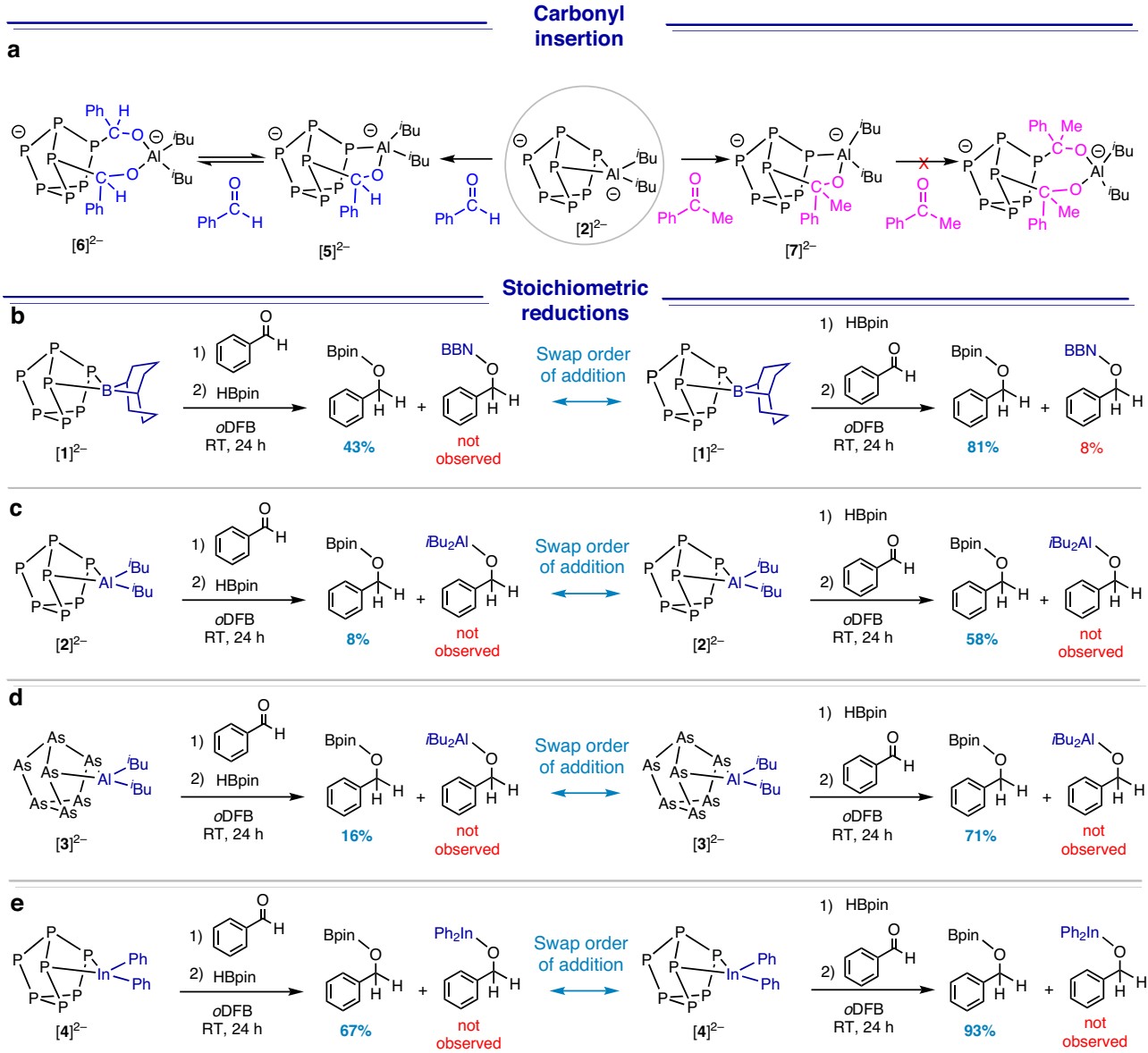

**Fig. 4 | Experimental mechanistic studies. a** Insertion of carbonyls into the P–Al bond in [**2**]²⁻. **b** Stoichiometric reduction of benzaldehyde using [**1**]²⁻. **c** Stoichiometric reduction of benzaldehyde using [**2**]²⁻. **d** stoichiometric reduction of benzaldehyde using [**3**]²⁻. **e** stoichiometric reduction of benzaldehyde using [**4**]²⁻.

energy change for the reaction, C₆H₅CHO + HBPin→C₆H₅CH₂O–BPin is −25.2 kcal/mol. The initial stages of the reaction involve the binding of one and then two benzaldehyde molecules to form the adducts [(Me₂Al)P₇]²⁻•C₆H₅CHO and [(Me₂Al)P₇]²⁻•2C₆H₅CHO discussed above (which correspond to the experimentally observed species, [**5**]²⁻ and [**6**]²⁻, respectively). The negligible free-energy change for the binding of the second benzaldehyde means that [(Me₂Al)P₇]²⁻•C₆H₅CHO and [(Me₂Al)P₇]²⁻•2C₆H₅CHO are in equilibrium, and an excess of HBpin can trap the former to generate the aluminium hydride intermediate **I1** through *trans*-metalation. In Fig. 5 **I1** is generated by the insertion of the B–H bond into the remaining Al–P bond of [(Me₂Al)P₇]²⁻•C₆H₅CHO, although it is also possible that it inserts into the Al–O bond, leaving the AlMe₂H unit anchored to the [P₇] cluster. With the hydride generated, a cycle based on the repeated insertion of C₆H₅CHO (ΔG = +5.2 kcal/mol) followed by *trans*-metalation with HBpin (ΔG = −30.4 kcal/mol) to release the product and regenerate the hydride. The sum of these final two steps is −25.2 kcal/mol, the overall free energy of the reaction. We note that this scheme represents an adaptation of the one in our previous paper dealing exclusively with

the boron catalyst, [**1**]²⁻, where we proposed that the HBpin inserted into the P–C bond in [**1**]²⁻•C₆H₅CHO rather than the E–P bond, as suggested in Fig. 5[34]. The detection of the double addition product, [**6**]²⁻, in the case of Al highlights the relative weakness of the remaining Al–P bond in [**5**] and this encouraged us to revise our original proposal, such that it is the E–P, rather than P–C bond that can break to accommodate the H[B] reducing agent.

Herein the synthesis of a family of group 13 functionalized [Pn₇] Zintl clusters is presented. This series of clusters was then systematically explored in their catalytic performance in mediating CO₂ reduction. In terms of TOFs, the performance decreases in the order: [κ²-(*i*Bu₂Al)P₇]²⁻ > [κ²-(*i*Bu₂Al)As₇]²⁻ > [κ²-(BBN)P₇]²⁻ > [κ²-(Ph₂In)P₇]²⁻. Determined to be the most active catalyst, under optimised conditions [Na(18c6)]₂[κ²-(*i*Bu₂Al)P₇] at 0.01 mol% gave a high TON of 9600, and high TOF of 741 h⁻¹ with 0.1 mol% at 60 °C. Comparison of [Na(18c6)]₂[κ²-(*i*Bu₂Al)P₇] with other homogenous main group catalyst under similar conditions in this transformation revealed it to be highly competitive, where it even outperformed numerous homogenous transition metal-based catalysts. The [Na(18c6)]₂[κ²-(*i*Bu₂Al)P₇] cluster

**Table 3 | Calculated free energies (kcal/mol) for the addition of one and two molecules of benzaldehyde to [X]²⁻ and selected bond lengths (Å)**

| | E–Pn (Å) | Pn–C (Å) | E–O (Å) | ΔG (kcal/mol) |
|---|---|---|---|---|
| [1]²⁻ | 2.09 | | | |
| [1]²⁻ + C₆H₅CHO→[1]²⁻•C₆H₅CHO | 2.11 | 1.95 | 1.48 | −11.5 |
| [1]²⁻•C₆H₅CHO + C₆H₅CHO→[1]²⁻· 2C₆H₅CHO | | 1.95 | 1.51 | +1.1 |
| [(Me₂Al)P₇]²⁻ | 2.45 | | | |
| [(Me₂Al)P₇]²⁻ + C₆H₅CHO→[(Me₂Al)P₇]²⁻•C₆H₅CHO | 2.44 | 1.96 | 1.79 | −13.6 |
| [(Me₂Al)P₇]²⁻•C₆H₅CHO + C₆H₅CHO→[(Me₂Al)P₇]²⁻•2C₆H₅CHO | | 1.96 | 1.79 | −1.8 |
| [(Me₂Al)As₇]²⁻ | 2.55 | | | |
| [(Me₂Al)As₇]²⁻ + C₆H₅CHO→[(Me₂Al)As₇]²⁻•C₆H₅CHO | 2.54 | 2.10 | 1.79 | −12.0 |
| [(Me₂Al)As₇]²⁻•C₆H₅CHO + C₆H₅CHO→[(Me₂Al)As₇]²⁻•2C₆H₅CHO | 2.63 | 2.11 | 1.79 | −2.7 |
| [4]²⁻ | 2.47 | | | |
| [4]²⁻ + C₆H₅CHO→[4]²⁻•C₆H₅CHO | 2.57 | 1.98 | 2.13 | +4.2 |
| [4]²⁻•C₆H₅CHO + C₆H₅CHO→[4]²⁻•2C₆H₅CHO | | 1.95 | 2.13 | +13.8 |
| [(Me₂Ga)P₇]²⁻ | 2.47 | | | |
| [(Me₂Ga)P₇]²⁻ + C₆H₅CHO→[(Me₂Ga)P₇]²⁻•C₆H₅CHO | 2.43 | 1.97 | 1.91 | −2.0 |
| [(Me₂Ga)P₇]²⁻•C₆H₅CHO + C₆H₅CHO→[(Me₂Ga)P₇]²⁻•2C₆H₅CHO | | 1.97 | 1.90 | +10.3 |

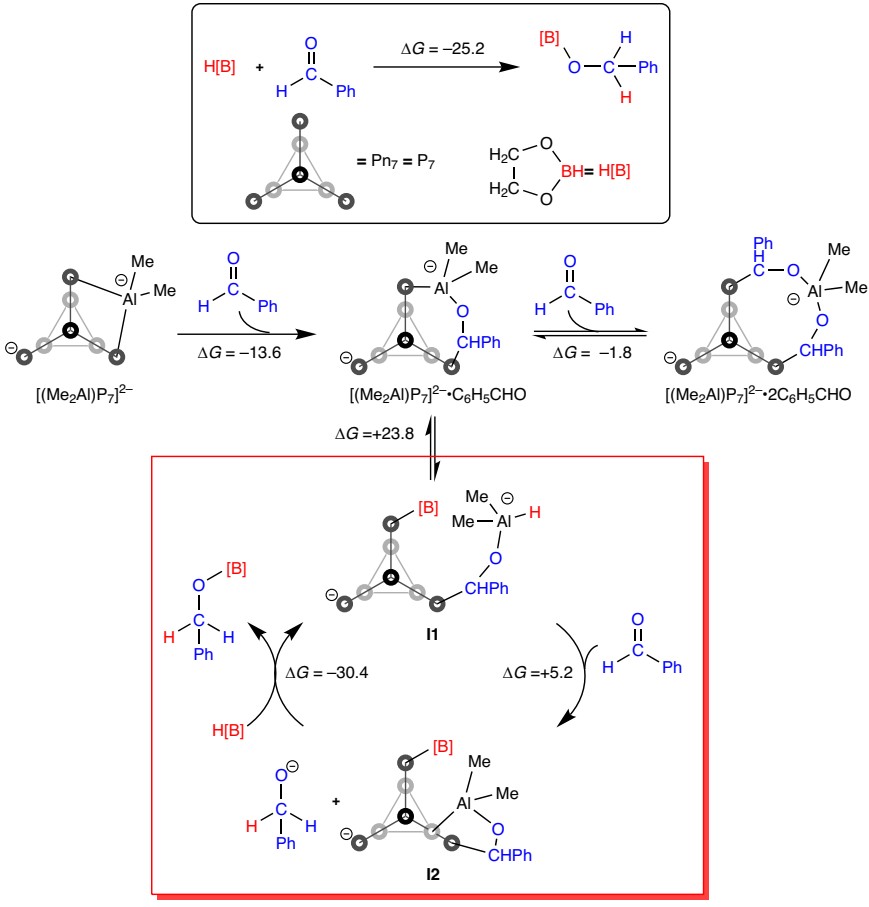

**Fig. 5 | Proposed catalytic cycle in which the bis-inserted carbonyl product [X]²⁻•2C₆H₅CHO is off-cycle.** The red box shows the catalytic cycle. ΔG change in Gibbs Free Energy. All energies are given in kcal/mol.

was found to be robust and could be recovered from the reaction mixture and recycled several times without loss in performance. Further experimental and computational investigations found a bis-carbonyl inserted species as an off-cycle product, and the determined kinetic isotope effect data was consistent with the H−B hydride being part of a slow step. The factors that direct the differences observed in catalytic competency are complex, and comprehensive mechanistic investigations are still ongoing. However, this work takes a detailed look at how tuning structural features at seven-atom pnictogen clusters modulates catalytic performance in $CO_2$ reduction chemistry, and further affirms the utility of transition metal-free Zintl clusters in important synthetic transformations.

## Methods

### Provided here are key protocols

Complete experimental details (general considerations, synthesis & characterization data, catalytic reactions, reactivity studies, kinetics measurements) and further computational details are provided in the Supplementary Information.

### Synthesis of [Na(18c6)]₂[κ²-(iBu₂Al)P₇] ([2]²⁻)

To a Schlenk flask charged with a stir bar and [Na(18c6)]₂[HP₇] (250 mg, 0.32 mmol, 1 equiv.), THF (5 mL) was added and cooled to −30 °C forming a slurry. A solution of diisopropyl aluminium hydride (45 mg, 0.32 mmol, 1 equiv.) in THF (5 mL) was cooled to −30 °C and dropwise added to the [Na(18c6)]₂[HP₇] slurry. Gas evolution was observed, and the reaction was allowed to react for 5 min at −30 °C, after which it was warmed to RT. The mixture was filtered yielding a

clear dark orange solution. The solvent was removed under reduced pressure, and the residue was washed with toluene (2 × 20 mL). The residue was dissolved in THF and filtered again, yielding a dark orange solution. Removal of volatiles under reduced pressure yielded glassy orange solids. Isolated Yield: 187 mg, 63%.

### Synthesis of [K(18c6)]₂[κ²-(iBu₂Al)As₇] ([3]²⁻)

To a Schlenk flask charged with a stir bar and [K(18c6)]₂[HAs₇] (250 mg, 0.22 mmol, 1 equiv.), THF (5 mL) was added and cooled to −30 °C forming a slurry. A solution of diisopropal aluminium hydride (31 mg, 0.22 mmol, 1 equiv.) in THF (5 mL) was cooled to −30 °C and dropwise added to the [K(18c6)]₂[HAs₇] slurry. Gas evolution was observed, and the reaction was allowed to react for 5 min at −30 °C, after which it was warmed to RT. The mixture was filtered yielding a black solution. The solvent was removed under reduced pressure, and the residue was washed with toluene (2 × 20 mL). The residue was dissolved in THF and filtered again, yielding a black solution. Removal of volatiles under reduced pressure yielded dark brown solids. Isolated Yield: 101 mg, 71%.

### General procedure for carbon dioxide hydroboration

To a J Young NMR tube C₆Me₆, HBBN dimer (36 mg, 0.15 mmol), and a solution of catalysts (with mol % relative to HBBN monomer) in oDFB:toluene (0.6 mL, 1:1) were added. The reaction mixture was degassed, and the headspace was refilled with $CO_2$ (1 atm). The reaction was monitored by ¹H, ¹¹B, and ¹¹B{¹H} NMR spectroscopy. The NMR conv. was calculated by integration of the crude ¹H NMR spectrum using the C₆Me₆ as an internal standard (¹H δ = 2.20 ppm).

## Computational methodology

All density functional theory calculations were performed using the ORCA 5.0.4 programme[52]. All calculations were performed using the r2SCAN-3c method[53], which is based on the r2SCAN functional[54], a bespoke mTZVPP basis set, and D4 and geometrical counterpoise (gCP) corrections[55,56]. The influence of the solvent was modelled using the CPCM model with the following parameters for oDFB: Dielectric Constant ($\varepsilon$) = 14.26 and Refractive Index ($n$) = 1.443[57,58]. All free energies include a concentration-induced correction of 1.89 kcal/mol to account for the change in standard state from gas phase (1 atm) to solution (1 mol/L)[59].

## Data availability

All data generated in this study are provided in Supplementary Information and Supplementary Data 1 and 2. The X-ray crystallographic coordinates for structures reported in this study have been deposited at the Cambridge Crystallographic Data Centre, under deposition numbers 2365860 ([Na(18c6)]$_2$[**2**]) and 2365861 ([K(18c6)]$_2$[**3**]). These data can be obtained free of charge from The Cambridge Crystallographic Data Centre via www.ccdc.cam.ac.uk/data_request/cif. All data are available from the corresponding author upon request.

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

## Acknowledgements

We thank the EPSRC and UKRI for funding (EP/V012061/1, EP/Y037391/1, and iCAT CDT EP/S023755/1) and supporting M.M., B.v.I., and W.D.J. We also thank Gareth Smith for mass spectrometric analyses, Anne Davies and Martin Jennings for elemental analyses, and Ralph Adams and Coral Mycroft for NMR spectroscopic enquiries. S.F.A. acknowledges the Saudi government for a postgraduate scholarship.

## Author contributions

B.v.I. performed all synthetic synthesis and kinetic work, and subsequent analysis and interpretation, with experiments designed in collaboration with M.M., B.v.I., and W.D.J. performed experimental mechanistic studies. S.F.A. and J.E.M. performed the computational mechanistic studies. B.v.I. and G.F.S.W. performed single crystal XRD analysis. B.v.I. wrote the initial drafts of the manuscript and supplementary information with contributions and edits from J.E.M. and M.M.

## Competing interests

The authors declare no competing interests.
