## [Transparent Peer Review file · Nature Communications]

Transforming Carbon Dioxide into a Methanol Surrogate using Modular Transition Metal-Free Zintl Ions

Corresponding Author: Professor Meera Mehta

Version 0:

Reviewer comments:

Reviewer #1

(Remarks to the Author)

The authors have discussed the transformation of Carbon Dioxide into a Methanol Surrogate by Zintl ions. They have explained the mechanism by doing both experiments and density functional theory-based calculations. The work presented in the manuscript is nice and well written. It shows the application of Zintl ions as catalyst in such transformation. Although the computational part has the supportive role, it requires attention to further strengthen the claim. The manuscript may be accepted for publication, but the authors have to perform a complete computational work.

1. In Table 3: The picture quality is not good. please provide the color code for the atoms.
2. The Grel is calculated directly or stepwise for 2C6H5CHO systems?
3. why Grel is positive for $[4]2 \rightarrow 2C6H5CHO$?
4. Provide all ground state geometries of the reactants, intermediates and products proposed in Figure 5 with energy and Free energy of the reaction.

Reviewer #2

(Remarks to the Author)

The manuscript entitled „Transforming Carbon Dioxide into a Methanol Surrogate using Modular Transition Metal-Free Zintl Ions” by Bono van IJzendoorn, Saad F. Albawardi, William D. Jobbins, George F. S. Whitehead, John E. McGrady, and Meera Mehta describes a decisive step forward on group 13 functionalised $[Pn7]3-$ ($Pn = P, As$) Zintl ions. They prepared a series of these compounds, including two unprecedented Al-functionalised Zintl clusters, and tested them in the catalytic hydroboration of CO₂ to a methanol surrogate (MeO-BBN). Intriguingly, one of the novel Al-containing species outperforms the groups previous B-based catalysts and even other main group as well as several transition metal species, competent in this reaction. Extensive experimental as well as computational investigations lead to a re-evaluation of their proposed catalytic cycle, which they now present in an updated and more detailed manner. The new compounds 2 and 3 are comprehensively characterized, and the overall quality of the manuscript is very high. This in mind, the manuscript certainly is of high interest for the broad readership of Nature Communications and qualifies for publication in this journal. However, there are some issues with the manuscript and the associated experimental work, which need to be considered before publication:

1. First and foremost, the absence of any kind of discussion of potential Ga-containing species throughout the entire manuscript is striking, as such species display a most obvious gap within the presented research. The authors should present convincing experimental evidence that such a species would follow the trend they propose, upon which an analogous Ga-species would be a less active catalyst in CO₂ hydroboration. If, during these studies, such a Ga-functionalised $[Pn7]3-$ cluster turns out to be inaccessible, impeding any catalytic studies, the authors should at least mention this within the manuscript, to clarify this issue to the reader.
2. In line 334 of the manuscript, the authors support their structural assignment for compound 5 and 6 by ¹H and ³¹P NMR spectroscopic data, without crystallographic structural proof. Of course it is understandable, that such crystallographic proof cannot always be obtained. However, additional spectroscopic data, namely ¹³C NMR and IR spectra, should be recorded in these cases and included into the discussion. Especially, the presence of ¹JC-H coupling in these compounds should simplify recording the respective ¹³C spectra.

3. Within the supporting information there is huge variation of the labelling of spectra. This should be made consistent and will then be easier to follow by the reader. For example, the assignment of NMR signals could be demonstrated in small insets showing the molecular structure of the compound(s).

4. The following smaller issues appear within the manuscript:

Line 17: The pnictogens are the nitrogen group elements.

Line 176: The labelling of TOFs should be made consistent throughout the manuscript, e. g. the term TOF50°C and TOFRT should be used.

Line 440: Conclusion instead of Discussion.

Reviewer #3

(Remarks to the Author)

In this paper, the authors have prepared a series of group 13 functionalized pnictogen clusters with the general formula $[(R_2E)Pn_7]_2^-$ (E = B, Al, In; Pn = P, As) and investigated their activity towards the hydroboration of CO₂. They found $[(iBu_2Al)P_7]_2^-$ was the optimal catalyst with high TONs and TOFs. However, I did not find the significance of this work and the quality cannot satisfy the Nature Communications. There are some problems lists as follows.

1. The novelty of this work should be further highlighted.
2. The isotope experiments should be conducted to reveal the origin of the produced products.
3. To confirm the reproducibility, the error bar should be added into Figure 3.
4. DFT calculations are suggested to be performed to reveal the possible reaction mechanism.

Version 1:

Reviewer comments:

Reviewer #1

(Remarks to the Author)

The authors made the necessary changes as per my queries. The revised manuscript may be suitable for possible publication in Nature Communications.

Reviewer #2

(Remarks to the Author)

After revision my concerns have been addressed and the paper is now ready for publication in Nature Communications.

Reviewer #3

(Remarks to the Author)

I have carefully checked the revised version. I think that the authors have made great improvements for this work.

RE: Revisions for Manuscript NCOMMS-24-39801

We thank all referees for their careful reviews of the manuscript, and are delighted with the highly positive assessment of the work. As per the decision letter, we have addressed all reviewer comments on a point-by-point basis in *blue* and with changes to the manuscript and supporting information highlighted in *yellow*.

Please note that due to the additional Figures and Tables added to the supporting information to address reviewer comments, all subsequent labelling has changed and this has not been highlighted for clarity. Further, references 51 and 59 in the manuscript are new, and this has altered subsequent reference numbers, which also has not been highlighted for clarity.

REVIEWER COMMENTS

Reviewer #1 (Remarks to the Author):

The authors have discussed the transformation of Carbon Dioxide into a Methanol Surrogate by Zintl ions. They have explained the mechanism by doing both experiments and density functional theory-based calculations. The work presented in the manuscript is nice and well written. It shows the application of Zintl ions as catalyst in such transformation. Although the computational part has the supportive role, it requires attention to further strengthen the claim. The manuscript may be accepted for publication, but the authors have to perform a complete computational work.

1. In Table 3: The picture quality is not good. please provide the color code for the atoms.

We agree that the pictures were not informative – we have decided to replace them with ChemDraw images which better highlight the important features.

2. The Grel is calculated directly or stepwise for 2C₆H₅CHO systems?

The Grel reported in the original version was calculated 'directly, as the difference between the sum of the energies of [X]²⁻ and two benzaldehyde molecules. This was stated in the original text (P 19). However, prompted by the reviewer's comment, we have decided to change this so that the numbers reported in the table now correspond to sequential binding energies, which fits better with the subsequent discussion of the mechanism. We also note here that the energies in the revised table are slightly different from those in the original. In the original submission, we had included a correction to the entropy change following the protocol proposed by Arai and Gellrich (PCCP, 2023, 25, 14005). As this makes only a minor contribution (a few kcal/mol at most), we have decided not to include it, and report the 'raw' free energy changes at 298.15 K, including the vibrational, rotational and translational entropies as well as the zero-point energies. This change does not impact our original conclusions from this work.

3. why Grel is positive for [4]2→•2C₆H₅CHO?

We believe that the reason for the positive Grel lies in the different strengths of the In–P vs

In–O bonds, compared to B–P/B–O and Al–P/Al–O. In is a rather softer metal which, in this context, means that the difference in intrinsic bond energy, $BE_{(E-P)}-BE_{(E-O)}$, is greater for In than it is for either Al or B. This correlates with the differences in bond lengths highlighted in the text. The binding of each molecule of aldehyde involves the formation of one E–O bond at the expense of one E–P bond (along with the formation of a P–C bond and the cleavage of a C=O π bond, which we view as approximately constant). Thus, the intrinsic strength of the In–P bond relative to In–O makes the binding less favourable, hence the positive ΔG^\ddagger . This point was discussed in the original text but is now given greater emphasis.

4. Provide all ground state geometries of the reactants, intermediates and products proposed in Figure 5 with energy and Free energy of the reaction.

The energetic data has been added to the cycle in Figure 5, along with a fuller discussion and a full free-energy surface is included in supporting information, Figure S119.

Reviewer #2 (Remarks to the Author):

The manuscript entitled „Transforming Carbon Dioxide into a Methanol Surrogate using Modular Transition Metal-Free Zintl Ions” by Bono van IJzendoorn, Saad F. Albawardi, William D. Jobbins, George F. S. Whitehead, John E. McGrady, and Meera Mehta describes a decisive step forward on group 13 functionalised $[Pn_7]^{3-}$ ($Pn = P, As$) Zintl ions. They prepared a series of these compounds, including two unprecedented Al-functionalised Zintl clusters, and tested them in the catalytic hydroboration of CO₂ to a methanol surrogate (MeO-BBN). Intriguingly, one of the novel Al-containing species outperforms the groups previous B-based catalysts and even other main group as well as several transition metal species, competent in this reaction. Extensive experimental as well as computational investigations lead to a re-evaluation of their proposed catalytic cycle, which they now present in an updated and more detailed manner. The new compounds 2 and 3 are comprehensively characterized, and the overall quality of the manuscript is very high. This in mind, the manuscript certainly is of high interest for the broad readership of Nature Communications and qualifies for publication in this journal. However, there are some issues with the manuscript and the associated experimental work, which need to be considered before publication:

1. First and foremost, the absence of any kind of discussion of potential Ga-containing species throughout the entire manuscript is striking, as such species display a most obvious gap within the presented research. The authors should present convincing experimental evidence that such a species would follow the trend they propose, upon which an analogous Ga-species would be a less active catalyst in CO₂ hydroboration. If, during these studies, such a Ga-functionalised $[Pn_7]^{3-}$ cluster turns out to be inaccessible, impeding any catalytic studies, the authors should at least mention this within the manuscript, to clarify this issue to the reader.

We agree with the reviewer that it would be ideal to complete the family by probing a Ga– Pn_7 cluster. However, we have been unable to obtain analytically pure material to accurately experimentally probe the catalytic competency of a Ga– Pn_7 cluster.

We have specified this in the revised manuscript (lines 116-120) and have added the relevant data in the SI (section 2.3.): “The synthesis of the related $[K^2-(Me_2N)_2Ga]P_7]^{2-}$ was also investigated by reacting $(Me_2N)_3Ga$ dimer with $[HP_7]^{2-}$ and resulted in ^{31}P NMR spectra consistent with the formation of the expected product. However, despite multiple efforts

isolation of analytically pure material was not achieved and precluded further catalytic investigations (see SI section 2.3.)."

We have also computed the free energies for binding of one and two molecules of benzaldehyde to a putative Ga-based catalyst, $[P_7GaMe_2]^{2-}$ (the Ga analogue of $[2]^{2-}$). The first binding energy of -0.1 kcal/mol is intermediate between those computed for Al (-13.6 kcal/mol) and In (+4.2 kcal/mol), and given that the first binding energy correlates with catalytic competency we would expect the reactivity of such a Ga-species to also be intermediary between the In- and Al-functionalized systems. Included in the manuscript lines 436-439.

2. In line 334 of the manuscript, the authors support their structural assignment for compound 5 and 6 by 1H and ^{31}P NMR spectroscopic data, without crystallographic structural proof. Of course it is understandable, that such crystallographic proof cannot always be obtained. However, additional spectroscopic data, namely ^{13}C NMR and IR spectra, should be recorded in these cases and included into the discussion. Especially, the presence of $^1J_{C-H}$ coupling in these compounds should simplify recording the respective ^{13}C spectra.

As requested by the reviewer we have conducted the IR experiments and these have been added to section 4.1.3. Unfortunately, the distinct C–O stretch of $[5]^{2-}$ and $[6]^{2-}$ are largely obscured by the 18c6 sequestering agent and only two shouldering signals could be observed labelled in Figure S41. The IR spectra are however consisted with the disappearance of the C=O stretch of benzaldehyde and the additional aryl stretches in the products. Further strong signals for aryl C–H bending can be observed in the fingerprint region for the products. Assignment of the IR spectra is discussed in the SI.

We have also conducted the ^{13}C NMR spectroscopic studies suggested, showing clear CH coupling compared to the $^{13}C\{^1H\}$ NMR spectrum. Further we have conducted ^{13}C DEPT135 NMR and $^1H^{13}C$ HSQC NMR spectroscopic experiments providing further evidence for the assignment of $[5]^{2-}$ and $[6]^{2-}$. All ^{13}C NMR recorded shows the distinctive $^1J_{CP}$ coupling as well, providing more support. All of these experiments are included in the SI in section 4.1.3.

*We have modified lines 348–349 with these new experiments added:
" 1H NMR, ^{13}C NMR, ^{31}P COSY NMR, and $^1H^{13}C$ HSQC (Heteronuclear Single Quantum Coherence) NMR spectroscopic studies are fully consistent with this proposal (see SI section 4.1.3.)."*

3. Within the supporting information there is huge variation of the labelling of spectra. This should be made consistent and will then be easier to follow by the reader. For example, the assignment of NMR signals could be demonstrated in small insets showing the molecular structure of the compound(s).

We agree with the reviewer and accordingly the labelling of the spectra in the SI have been altered. Where appropriate compound structures have been added into or above the spectra with colour coded dots to aid readability. We endeavoured to have full consistency throughout the SI to make everything easy to follow for the reader, however due to the nature of some spectra and/or extra information being identified, there are some minor labelling variations, e.g. the ^{31}P COSY NMR spectra (Figure S39) P atoms are numbered as coloured dots would not have been helpful in this context.

As almost all spectra have been slightly altered to make their labelling more consistent, this

has not been highlighted in the ESI for clarity.

4. The following smaller issues appear within the manuscript:

Line 17: The pnictogens are the nitrogen group elements.

Line 176: The labelling of TOFs should be made consistent throughout the manuscript, e. g. the term TOF50°C and TOFRT should be used.

Line 440: Conclusion instead of Discussion.

We agree with the reviewer, and have adapted the manuscript accordingly.

Reviewer #3 (Remarks to the Author):

In this paper, the authors have prepared a series of group 13 functionalized pnictogen clusters with the general formula $[(R_2E)Pn_7]^{2-}$ (E = B, Al, In; Pn = P, As) and investigated their activity towards the hydroboration of CO₂. They found $[(iBu_2Al)P_7]^{2-}$ was the optimal catalyst with high TONs and TOFs. However, I did not find the significance of this work and the quality cannot satisfy the Nature Communications. There are some problems lists as follows.

1. The novelty of this work should be further highlighted.

In order to better highlight the novelty of this work to the reader, we outlined several key findings from this work in the last paragraph of the introduction, lines 97-104.

2. The isotope experiments should be conducted to reveal the origin of the produced products.

We have conducted ¹³C-labelled experiments using ¹³CO₂. The results show the ¹³CO₂ gas to be the origin of the observed products, most indicative the ¹J_{HC} coupling now observed in the ¹H NMR spectra. The following sentence has been added to the manuscript (line 175-176) and corresponding data has been added in the SI (section 3.2.):

“Isotopic labelled studies using ¹³C-labelled CO₂ confirmed the origin of the carbon products (see SI 3.2).”

3. To confirm the reproducibility, the err bar should be added into Figure 3.

We have conducted more experiments to provide additional support for the reproducibility of the recyclability of catalysts $[Na(18c6)]_2[iBu_2AlP_7]$. The 95% confidence interval has been determined and error bars have been added to Figure 3. The SI has been adapted accordingly, see Section 3.4 table S1 and Section 3.5.

4. DFT calculations are suggested to be performed to reveal the possible reaction mechanism.

In line with the suggestion of this referee and also of referee 1, we have replaced the schematic mechanistic proposal in Figure 5 with a free-energy diagram. We have, in fact, conducted a very extensive set of calculations on possible mechanistic pathways, the totality of which is far beyond the scope of the current paper, where the emphasis is primarily on the experimental results. We have considered possible mechanisms where the P–P bonds are

cleaved in intermediates, as well as those where the P_7 cluster is effectively 'inert', acting as a scaffold for the important transformations. This full mechanistic study will be the subject of a separate publication. We emphasise that DFT calculations cannot, in our view, 'reveal' the mechanism – all that is ever possible is to test various mechanistic hypotheses and eliminate those that are inconsistent with the data, usually because the calculated barriers to fundamental steps are too high. The data shown in Figure 5 represent our 'best' model i.e. the one that is most consistent with the data.